# Activity driven transport in harmonic chains

**Ion Santra[1] and Urna Basu[1,2][⋆]**

**1** Raman Research Institute, C. V. Raman Avenue, Bengaluru 560080, India
**2** S. N. Bose National Centre for Basic Sciences, JD Block, Saltlake, Kolkata 700106, India

⋆ urna@bose.res.in

## Abstract

The transport properties of an extended system driven by active reservoirs is an issue of paramount importance, which remains virtually unexplored. Here we address this issue, for the first time, in the context of energy transport between two active reservoirs connected by a chain of harmonic oscillators. The couplings to the active reservoirs, which exert correlated stochastic forces on the boundary oscillators, lead to fascinating behavior of the energy current and kinetic temperature profile even for this linear system. We analytically show that the stationary active current (i) changes non-monotonically as the activity of the reservoirs are changed, leading to a negative differential conductivity (NDC), and (ii) exhibits an unexpected direction reversal at some finite value of the activity drive. The origin of this NDC is traced back to the Lorentzian frequency spectrum of the active reservoirs. We provide another physical insight to the NDC using nonequilibrium linear response formalism for the example of a dichotomous active force. We also show that despite an apparent similarity of the kinetic temperature profile to the thermally driven scenario, no effective thermal picture can be consistently built in general. However, such a picture emerges in the small activity limit, where many of the well-known results are recovered.

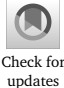

# 1 Introduction

Understanding energy transport properties of driven systems is a central issue of nonequilibrium statistical physics. Theoretical attempts in this regard often rely on the study of simple, yet analytically tractable model systems [1, 2]. A paradigmatic example is a chain of harmonic oscillators connected to thermal reservoirs of different temperatures at the two ends, first studied by Rieder, Lieb, and Lebowitz (RLL) in a seminal work [3]. They showed that this system reaches a nonequilibrium stationary state carrying a thermal current, which survives in the thermodynamic limit. Several generalizations of this simple model have been studied by introducing disorder, anharmonic interactions, pinning potentials and activity in the bulk [4–11]. In almost all of these studies, however, the reservoirs attached to the system are taken to be equilibrium ones — the random and dissipative forces exerted by each reservoir on the boundary oscillators satisfy the Fluctuation-Dissipation theorem (FDT) [12].

Nonequilibrium reservoirs, on the other hand, do not respect any such FDT, giving rise to a wide range of new possibilities [13–17]. For example, energy transport in systems connected to nonequilibrium reservoirs show non-monotonic kinetic temperature profile, negative differential thermal conductivity and non-reciprocal heat transport [18–21]. Active reservoirs refer to a special class of nonequilibrium reservoirs, consisting of self-propelled particles like bacteria or Janus beads, which are inherently out of equilibrium by consuming energy from the environment at an individual level [22–24]. Recent studies, both theoretical and experimental, show that individual probe particles immersed in such active reservoirs exhibit many unusual features including emergence of negative friction, modification of equipartition theorem and anomalous relaxation dynamics [26–35]. A natural question is how the transport properties of an extended system are affected when connected to active reservoirs at the boundaries. To the best of our knowledge, this has not been studied so far.

In this article, we ask this question in a simple setting similar to RLL model—an ordered chain of harmonic oscillators connected to two active reservoirs at the two ends. The active reservoirs exert stochastic forces on the boundary oscillators, which do not satisfy FDT. As a simple model, we consider that this stochastic force has an exponentially decaying autocorrelation, which is a common feature of active dynamics, the autocorrelation time-scale being a measure of the activity of the reservoirs. In the long-time limit the system reaches a nonequilibrium stationary state (NESS) carrying an energy current which we compute exactly. We find that this current shows two remarkable features, namely, an unexpected direction reversal and a negative differential conductivity (NDC) whose origin lies in the Lorentzian frequency spectra of the active reservoirs. The emergence of the NDC and current reversal in a linear system without any kinetic constraints sets it apart from the few similar phenomena observed previously [18, 36–40]. For a specific model of a dichotomous active force, we illustrate that the NDC can also be viewed as a result of a positive correlation of the current and the number of directional flips of the force. We also show that the kinetic temperature profile retains strong



Figure 1: Schematic representation of a chain of oscillators connected to two nonequilibrium reservoirs which exert active forces $f_{1,N}(t)$ on the boundary oscillators.

signatures of activity despite attaining a uniform value in the bulk. In the limit of small activity, the reservoirs behave somewhat similar to thermal ones and the well-known properties of RLL-model are recovered.

## 2 Model

We consider a one-dimensional chain with $N$ particles, each with mass $m$, connected by harmonic springs of stiffness $k$, attached to two different active reservoirs at the boundaries [see Fig. 1]. The coupling to the active reservoir is modeled by including a stochastic force on the boundary particle, in addition to the usual dissipative and white-noise forces coming from an equilibrium thermal reservoir. The equations of motion for $x_l$, the displacement of the $l$-th particle from its equilibrium position, read,

$$m\ddot{x}_1 = -k(2x_1 - x_2) - \gamma \dot{x}_1 + \xi_1(t) + f_1(t), \tag{1a}$$

$$m\ddot{x}_l = -k(2x_l - x_{l-1} - x_{l+1}), \quad \forall \, l \in [2, N-1], \tag{1b}$$

$$m\ddot{x}_N = -k(2x_N - x_{N-1}) - \gamma \dot{x}_N + \xi_N(t) + f_N(t), \tag{1c}$$

where we have used fixed boundary conditions $x_0 = 0 = x_{N+1}$. We assume that the thermal components of the reservoirs are at temperatures $T_1$ and $T_N$, so that the white noises $\xi_{1,N}(t)$ acting on the boundary particles are related to the dissipation $\gamma$ through FDT,

$$\langle \xi_l(t)\xi_j(t') \rangle = 2\gamma T_j \delta_{l,j} \delta(t-t') \quad \text{where} \quad j, l = 1, N. \tag{2}$$

The FDT is violated by the presence of the active forces $f_{1,N}(t)$ which are assumed to be independent stationary colored noises. Most commonly, such active noises have an exponentially decaying correlation, $\langle f_j(t) f_l(t') \rangle = \delta_{jl} a_j^2 \exp(-|t - t'|/\tau_j)$, where $a_j$ denotes the strength of the noise and the correlation-time $\tau_j$ is a measure of the activity. As a specific example, we consider the dichotomous noise

$$f_j(t) = a_j \sigma_j(t), \tag{3}$$

where $\sigma_j$ alternates between $\pm 1$ at a constant rate $\alpha_j$, giving rise to an exponential correlation with $\tau_j = 1/(2\alpha_j)$. However, our main results remain quite robust for general active driving, since exponential correlations generically appear in active processes including run-and-tumble motion, active Brownian motion and direction reversing active Brownian motion [42–44].

## 3 Results

We first present a brief summary of our main results. The primary observables of interest here are the energy current and the kinetic temperature profile, both of which we compute exactly.

The energy current flowing through the system is most conveniently expressed as [2],

$$J \equiv \langle \mathcal{J}(t) \rangle = \left\langle \dot{x}_1[-\gamma \dot{x}_1 + \xi_1(t) + f_1(t)] \right\rangle, \tag{4}$$

where the average is taken in the NESS. Because of the linear nature of the equations of motion the stationary current naturally separates into two components, an *active* one induced by the activity driving and a thermal one proportional to the temperature difference of the two reservoirs (same as in the usual RLL setup [3], which is quoted in Eq. (15)). We show that the active current in the thermodynamic limit is given by,

$$J_{\text{act}} = \frac{m}{2\gamma^2}(a_1^2 \mathcal{E}_1 - a_N^2 \mathcal{E}_N), \quad \text{with} \tag{5}$$

$$\mathcal{E}_j = \frac{\tau_j^2 k^2 \left[ \sqrt{1 + \frac{4\gamma^2}{mk}} - 1 \right] + \gamma^2 \left[ 1 - \sqrt{1 + \frac{4k\tau_j^2}{m}} \right]}{2\tau_j(\tau_j^2 k^2 - \gamma^2)}. \tag{6}$$

There are a number of striking features of this active current which distinguishes it from the usual thermal current. First, $J_{\text{act}}$ exhibits a non-monotonic behavior as the activity of either of the reservoirs is changed, giving rise to a negative differential conductivity [see Fig. 2]. More surprisingly, the current reverses its direction as the activity of one of the reservoirs, say $\tau_1$, is changed at a non-trivial value $\tau_1^* \neq \tau_N$ [see the phase diagram in Fig. 4].

We also show that the stationary kinetic temperature profile $\hat{T}_l = m\langle \dot{x}_l^2 \rangle$ attains a constant value in the bulk $1 \ll l \ll N$ with an exponentially decaying boundary layer. Surprisingly, we find that, the bulk temperature can be expressed in a form similar to the famous RLL result [3],

$$\hat{T}_{bulk} = \frac{1}{2}(\mathcal{T}_1 + \mathcal{T}_N), \quad \text{with} \quad \mathcal{T}_j = \frac{a_j^2 \tau_j}{\gamma \sqrt{1 + 4\tau_j^2 k/m}}. \tag{7}$$

This would suggest the possibility of interpreting $\mathcal{T}_{1,N}$ as 'effective temperatures' associated to the two active reservoirs. However, we show that such an interpretation is not acceptable and the active reservoirs remain essentially different from thermal ones.

In the limit of small activity $\tau_1, \tau_N \ll \sqrt{m/k}$, however, an effective thermal picture emerges. In this case, we show that, the active forces behave somewhat similar to white noises and the energy current and bulk kinetic temperature are consistent with the system being connected to thermal reservoirs with effective temepartures $T_j^{\text{eff}} = T_j + a_j^2 \tau_j/\gamma$. However, the signatures of activity still remain in some atypical features, like the presence of a non-trivial boundary layer even when $T_1^{\text{eff}} = T_N^{\text{eff}}$.

We start by rewriting Eqs. (1) as,

$$M\ddot{X}(t) = -\Phi X(t) - \Gamma \dot{X}(t) + \Xi(t) + F(t), \tag{8}$$

where $X(t) = \{x_l(t); l = 1, \ldots, N\}$ is a vector and $M$ is an $N$-dimensional diagonal matrix with $M_{lj} = m\delta_{l,j}$. Moreover, $\Gamma$ and $\Phi$ are $N$-dimensional matrices given by

$$\Gamma_{lj} = \gamma(\delta_{l,1}\delta_{j,1} + \delta_{l,N}\delta_{j,N}), \Phi_{lj} = k\left(2\delta_{l,j} - \delta_{l,j-1} - \delta_{l,j+1}\right).$$

Finally, $\Xi_j = \xi_1(t)\delta_{j1} + \xi_N(t)\delta_{jN}$ and $F_j = f_1(t)\delta_{j1} + f_N(t)\delta_{jN}$ are vectors.

We are interested in the solution of Eq. (8) in the stationary state, which is most conveniently obtained by taking a Fourier transform with respect to time, $\tilde{X}(\omega) = \int_{-\infty}^{\infty} dt\, e^{i\omega t} X(t)$. This leads to,

$$\tilde{X}(\omega) = G(\omega)\left[\tilde{\Xi}(\omega) + \tilde{F}(\omega)\right], \tag{9}$$

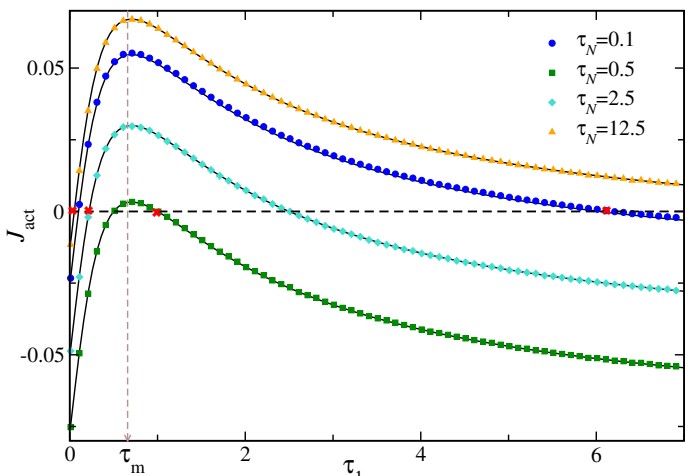

Figure 2: Activity induced current $J_{\text{act}}$ *vs* $\tau_1$ for different values of $\tau_N$ and $\gamma = 1 = m = a_1 = a_N$ and $k = 2$; symbols indicate the data obtained from numerical simulations with $N = 64$ oscillators and the solid black lines indicate the analytical prediction Eq. (6). The red crosses mark the non-trivial current reversal points and $\tau_m$ indicates the value of the activity for $J_{\text{act}}$ is maximum.

where $\tilde{\Xi}(\omega)$ and $\tilde{F}(\omega)$ are the Fourier transforms of $\Xi(t)$ and $F(t)$ respectively and

$$
\begin{aligned}
G(\omega) &= [-M\omega^2 + \Phi - i\omega(\Gamma_L + \Gamma_R)]^{-1} \\
&= \begin{bmatrix} -m\omega^2 + 2k - i\omega\gamma & -k & 0 & \cdots \\ -k & -m\omega^2 + 2k & -k & \cdots \\ \vdots & \vdots & \ddots & \cdots \\ 0 & \cdots & -k & -m\omega^2 + 2k - i\omega\gamma \end{bmatrix}^{-1}.
\end{aligned} \tag{10}
$$

Inverting the transform, we get from Eq. (9),

$$
X(t) = \frac{1}{2\pi} \int_{-\infty}^{\infty} d\omega\, e^{-i\omega t} G(\omega) \left[ \tilde{\Xi} + \tilde{F} \right]. \tag{11}
$$

To compute the steady state energy current $J$ defined in Eq. (4), we need the autocorrelation of the stochastic forces $\xi_j(t)$ and $f_j(t)$ in the Fourier-space,

$$
\begin{aligned}
\langle \tilde{\xi}_j(\omega)\tilde{\xi}_l(\omega') \rangle &= 4\pi\gamma T_j \delta_{jl}\delta(\omega + \omega'), \tag{12a} \\
\langle \tilde{f}_j(\omega)\tilde{f}_l(\omega') \rangle &= 2\pi\delta_{jl}\delta(\omega + \omega')\tilde{g}(\tau_j, \omega). \tag{12b}
\end{aligned}
$$

Here $\tilde{g}(\tau_j, \omega) = \frac{2a_j^2 \tau_j}{1 + \omega^2\tau_j^2}$ denotes the spectral density of the active force from the $j$th reservoir, which clearly is a Lorentzian with corner frequency $\tau_j^{-1}$.

## 3.1 Stationary energy current

The independence of the thermal and active noises along with the linear nature of the couplings lead to the current in Eq. (4) to separate into two components $J = J_{\text{th}} + J_{\text{act}}$; see Appendix. A for details. The thermal current, generated due to the temperature gradient,

$$
J_{\text{th}} = \gamma^2(T_1 - T_N) \int_0^{\infty} \frac{d\omega}{\pi} \omega^2 |G_{1N}(\omega)|^2, \tag{13}
$$

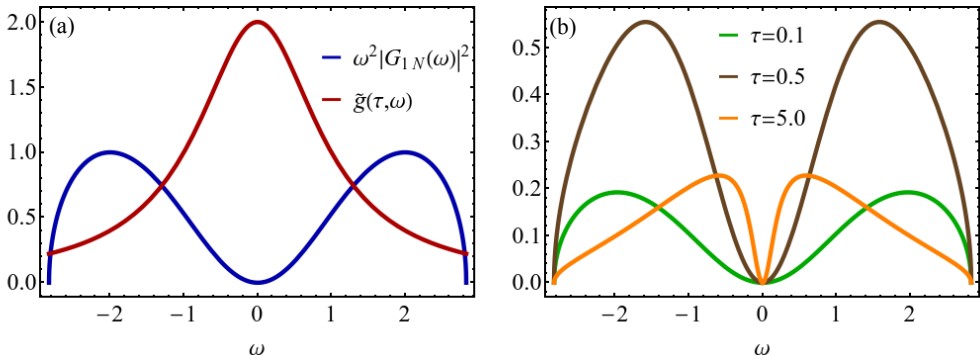

Figure 3: (a) Plot of the phonon transmission coeffcient $\omega^2|G_{1N}(\omega)|^2$ in the $N \to \infty$ limit [see Eq. (A.29)] and the reservoir spectrum $\tilde{g}(\tau, \omega)$ [see Eq. (A.11)] for $\tau = 0.5$ as functions of $\omega$. (b) Plot of the single reservoir transmission coefficient $\omega^2|G_{1N}(\omega)|^2\tilde{g}(\tau, \omega)$ *vs* $\omega$ for different values of $\tau$. Here we have taken $m = 1, k = 2$ and $\gamma = 1$.

remains the same as in the case of equilibrium reservoirs and can be computed explicitly [3,6]. The active nature of the reservoirs gives rise to the additional current,

$$J_{\text{act}} = \gamma \int_0^\infty \frac{d\omega}{\pi} \omega^2|G_{1N}(\omega)|^2 \Big[ \tilde{g}(\tau_1, \omega) - \tilde{g}(\tau_N, \omega) \Big], \tag{14}$$

where $\tilde{g}(\tau_j, \omega)$ contains information about the reservoir activity. Equation (14) is a Landauer-like formula, where the Lorentzian reservoir spectra $\tilde{g}(\tau_j, \omega)$ couples to the phonon transmission coefficient $\omega^2|G_{1N}(\omega)|^2$.

To compute the currents explicitly we need $G_{1N}(\omega)$, which is obtained by exploiting the tridiagonal structure of $G^{-1}(\omega)$ [5, 6, 10, 45]. We are particularly interested in the thermodynamic limit $N \to \infty$, where $G_{1N}(\omega)$ vanishes exponentially outside the phonon band $|\omega| > \omega_c = 2\sqrt{k/m}$ [5]. In that limit, we show that, the contribution from the $j$-th reservoir $(j = 1, N)$ is given by [see Appendix. A for details],

$$\gamma \int_0^\infty \frac{d\omega}{\pi} \omega^2|G_{1N}(\omega)|^2\tilde{g}(\tau_j, \omega) = \int_0^\pi \frac{dq}{\pi} \frac{mka_j^2\tau_j \sin^2 q}{[mk + 2\gamma^2(1 - \cos q)][m + 2k\tau_j^2(1 - \cos q)]}, \tag{15}$$

where $\omega$ and $q$ are related by $m\omega^2 = 2k(1 - \cos q)$. Computing the $q$-integral and combining the contributions from both the reservoirs, we get the active current flowing through the system in the thermodynamic limit which is quoted in Eq. (6).

Figure 2 shows a plot of the predicted $J_{\text{act}}$ as a function of the left reservoir activity $\tau_1$ for a set of different values of $\tau_N$. This shows an excellent match with the current measured from numerical simulations with a chain of oscillators driven by the dichotomous noise given in Eq. (3). The figure illustrates some remarkable features of the active current which we discuss below.

### 3.1.1 Negative differential conductivity

The active current shows a non-monotonic behavior—as $\tau_1$ is increased, $J_{\text{act}}$ initially increases until reaching a maximum value after which it starts to decrease. It is clear from Eq. (6) that this non-monotonic behavior is inherent to the individual contributions from both the reservoirs — if $\tau_N$ is increased, keeping $\tau_1$ fixed, a similar behavior is seen where the current first decreases and then starts to increase. The existence of this non-monotonic behavior becomes qualitatively clear by looking at the frequency spectrum of the reservoir $\tilde{g}(\tau, \omega)$. From

Eq. (12b), it is clear that $\tilde{g}(\tau,\omega)$ is a Lorentzian, peaked around $\omega = 0$ with width $\sim \tau^{-1}$. On the other hand, the phonon transmission coefficient $\omega^2 |G_{1N}(\omega)|^2$ is peaked around the characteristic frequency $\omega_c = 2\sqrt{k/m}$, with a minimum at $\omega = 0$ [see Fig. 3(a)]. Consequently, the overlap of the system transmission coefficient and the reservoir spectra changes non-monotonically as $\tau$ is changed, reaching a maximum at some intermediate value of $\tau^{-1} \in [0, \omega_c]$ [see Fig. 3(b)]. This, in turn, gives rise to the non-monotonic behavior of $J_{\text{act}}$, which shows a maximum (minimum) as $\tau_1$ ($\tau_N$) is varied. In fact, it can be easily seen from Eq. (6) that for large $k$, the current is maximum at a value of $\tau_1 = \tau_m \propto \omega_c^{-1}$.

The non-monotonic behavior implies that the differential conductivity $\chi_j = \frac{dJ_{\text{act}}}{d\tau_j}$, which is nothing but the linear response of the current to a small change in the activity of the $j$-th reservoir, becomes negative in some parameter regimes. Noneqilibrium response theory provides a way to express this coefficient in terms of correlations of some physical observables [46, 47]. For the simple dynamics (3), using a trajectory based approach, we find [see Appendix C],

$$\chi_j = \lim_{t \to \infty} -\frac{1}{\tau_j} [\langle n_j(t) \mathcal{J}(t)\rangle - \langle n_j(t)\rangle\langle \mathcal{J}(t)\rangle], \tag{16}$$

where $n_j(t)$ denotes the total number of flips of $\sigma_j$ during a time interval $[0, t]$, $\mathcal{J}(t)$ is the instantaneous current and the average is computed in the unperturbed system. The above equation implies that when the number of flips $n_j(t)$ is positively correlated with the current an NDC emerges.

### 3.1.2 Current reversal

There is another, more striking, behavior induced by the presence of the active driving, namely, reversal of the direction of the current. We see from Fig. 2, that for any given $\tau_1$, $J_{\text{act}}$ reverses its direction twice—once (trivially) at $\tau_1 = \tau_N$ and again at another value $\tau_1 = \tau_1^*$ which depends non-trivially on $\tau_N$. For a fixed $\tau_N$, $J_{\text{act}}$ begins with a negative value (energy flowing from right to left reservoir) for $\tau_1 = 0$, which becomes positive (energy flowing from left to right reservoir) with increase in $\tau_1$. However, on increasing $\tau_1$ further, the current again reverses its direction and becomes negative. Mathematically, this additional reversal can be understood from the observation that for a fixed value of $\tau_N$, $\mathcal{E}_1 \to 0$ for both $\tau_1 \to 0$ and $\tau_1 \to \infty$ [see Eq. (6)], and consequently $J_{\text{act}}$ has the same negative value at these two limits. Now, since $J_{\text{act}}$ must reverse sign at $\tau_1 = \tau_N$, an additional reversal is required to reach the limiting negative values. A similar scenario is observed when $\tau_N$ is changed keeping $\tau_1$ fixed, as expected from the symmetry of the system.

This behavior is illustrated in Fig. 4; panel (a) shows a three-dimensional plot of $J_{\text{act}}$ on the $(\tau_1, \tau_N)$ plane, while Fig. 4(b) shows the two-dimensional projection of (a) indicating the regions $J_{\text{act}} > 0$ and $J_{\text{act}} < 0$. For any given $\tau_N$, the current reverses its direction at $\tau_1 = \tau_N$ and another non-trivial point $\tau_1 = \tau_1^*(\tau_N)$. The latter is given by the non-trivial solution of $a_1^2 \mathcal{E}_1(\tau_1) = a_N^2 \mathcal{E}_N(\tau_N)$. Similarly, for any given $\tau_1$, the current reversal occurs at $\tau_N = \tau_1$ and $\tau_N^*(\tau_1)$ [indicated by the solid red curve in 4(b)]. Interestingly, the intersection of the curves $\tau_1 = \tau_N$ and $\tau_1 = \tau_1^*(\tau_N)$ denoted by $\tau_1 = \tau_N = \bar{\tau}$ is a saddle point, as can be seen from Fig. 4(a). The current does not change direction when one passes through the saddle point—for $\tau_N = \bar{\tau}$, the current remains negative for all values of $\tau_1 \neq \tau_N$, while for $\tau_1 = \bar{\tau}$, the current remains positive for all values of $\tau_N \neq \tau_1$.

NDC and current reversal have been observed in certain nonequilibrium systems with non-linearity, presence of obstacles or kinetic constraints [18, 36–40]. Surprisingly, the dynamical active driving here gives rise to both features even in a linear chain.

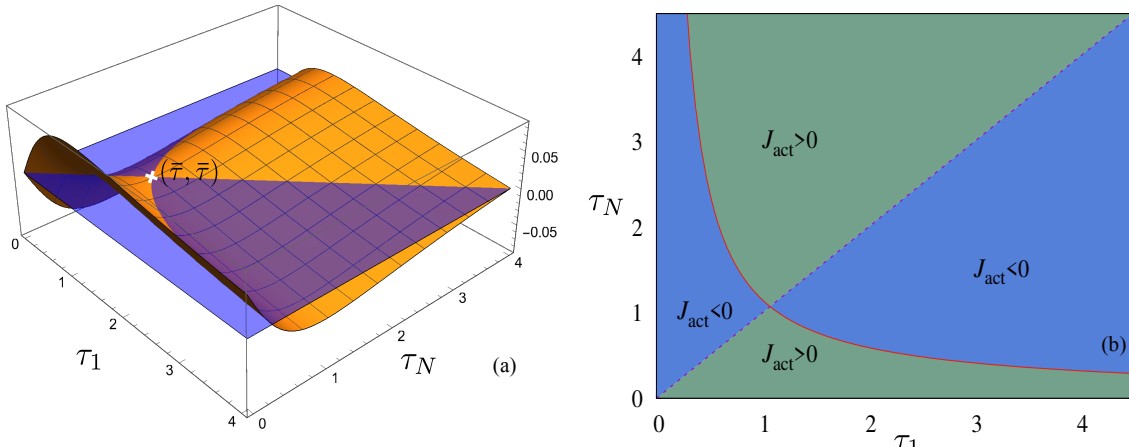

Figure 4: (a) Three dimensional plot of $J_{\text{act}}$ on the $(\tau_1, \tau_N)$ plane: the meshed orange surface denotes $J_{\text{act}}$ given by Eq. (5) in the main text, while the un-meshed, semi-transparent blue surface corresponds to $J_{\text{act}} = 0$. The curves formed by intersection of these two surfaces give the locus of the zeros of the active current, the intersection of these two curves are denoted by $\tau_1 = \tau_N = \bar{\tau}$, which clearly is a saddle point of $J_{\text{act}}$. $J_{\text{act}} > 0$ in the region above the blue surface, while $J_{\text{act}} < 0$ in the region below the blue surface. (b) Two dimensional projection of (a) showing phase diagram of $J_{\text{act}}$ in the $(\tau_1, \tau_N)$ plane—: The light blue (green) shade indicates the region where the active current is negative (positive). The continuous red curve shows $\tau_N^*$ as a function of $\tau_1$ whereas the dashed curve indicates the line $\tau_N = \tau_1$.

## 3.2 Kinetic temperature

The average kinetic energy of the oscillators provides a way to define a local 'temperature' for driven oscillator chains [1,3]. For purely thermal drive, this kinetic temperature is uniform in the bulk of the system and is given simply by $(T_1 + T_N)/2$ in the $N \to \infty$ limit. Here we are interested in the effect of the active drive on the kinetic temperature $\hat{T}_l = m\langle \dot{x}_l^2(t)\rangle$ and thus consider $T_1 = T_N = 0$. In this case, using Eq. (11), we get,

$$\hat{T}_l = m \int \frac{d\omega}{2\pi} \omega^2 \Big[ |G_{l1}(\omega)|^2 \tilde{g}(\tau_1, \omega) + |G_{lN}(\omega)|^2 \tilde{g}(\tau_N, \omega) \Big]. \tag{17}$$

The matrix elements can again be computed exploiting the tridiagonal structure of $G^{-1}(\omega)$. Performing a similar calculation as before [see Appendix. B for details], we find that, in the thermodynamic limit $N \to \infty$, the steady state temperature profile is flat in the bulk, accompanied by exponentially decaying boundary layers. The bulk temperature $\hat{T}_{\text{bulk}}$ can be obtained explicitly and is quoted in Eq. (7). The predicted value of bulk temperatures for a fixed $\tau_1$ and different values of $\tau_N$ are plotted in Fig. 5(a) along with numerical simulations performed with the active force given in Eq. (3); the excellent agreement validates our prediction. Interestingly, boundary kinks in the $\hat{T}_l$ profile, which are generically present for coupling with thermal reservoirs [49], are absent here.

The form of Eq. (7) raises a possibility of associating an effective temperature $\mathcal{T}_j$ to the $j$-th active reservoir. At first glance, this identification also appears to be consistent with a 'zeroth law' — when $\tau_1 = \tau_N$, i.e., $\mathcal{T}_1 = \mathcal{T}_N$, the bulk of the system is at the same 'temperature' as the reservoirs. However, such an interpretation is not acceptable for several reasons. First, note that the kinetic temperatures of the boundary sites $\hat{T}_{1,N}$ remain different from $\hat{T}_{\text{bulk}}$ giving rise to a boundary layer even when $\tau_1 = \tau_N$ [see Fig. 5(a)] which is absent for ordinary equilibrium reservoirs. Moreover, the stationary active current (6) is very different than the

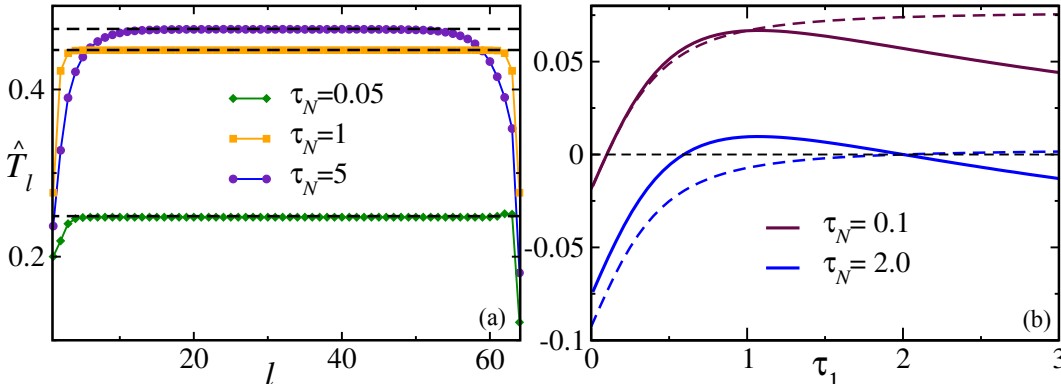

Figure 5: (a) The kinectic temperature profile $\hat{T}_l$ for $\tau_1 = 1$ and different values of $\tau_N$ measured from simulations with a chain of $N = 64$ oscillators driven by the active noises (3). The dashed black lines show the predicted bulk temperature (7). (b) Comparison of $J_{\text{act}}$ (solid lines) with the expected current for 'effective' temperature gradient $\mathcal{T}_1 - \mathcal{T}_N$ (dashed lines) for two different values of $\tau_N$. Here $m = 1, k = 1, \gamma = 1$ and $a_1 = a_N = 1$.

energy current which would have been generated if the system were connected to thermal reservoirs of temperatures $\mathcal{T}_1$ and $\mathcal{T}_N$ at the two ends. This is illustrated in Fig. 5(b) which shows neither current reversal nor any NDC in the 'effective' thermal scenario. However, the effective temperature picture becomes viable in the limit of small activity, which we discuss next.

**Passive limit-** It is well known that active systems show an effective passive behavior in the limit of vanishing correlation time [42–44]. Similarly, in our case, when $\tau_j \to 0$, the active force $f_j(t)$ resembles a white noise with effective correlation $\langle f_j(t)f_j(t')\rangle \to a_j^2\tau_j\delta(t - t')$. In this limit, the active forces in Langevin Eqs. (1) can be thought of representing thermal reservoirs with effective temperatures $a_j^2\tau_j/\gamma$ and satisfying FDT. The well known results of the RLL model are expected to be recovered in this 'thermal' limit. Indeed we see from Eq. (7) that when the active time-scales are much smaller than the coupling time-scale, i.e., $\tau_1, \tau_N \ll \sqrt{m/k}$, the kinetic temperature associated with the reservoirs $T_j^{\text{eff}} \simeq a_j^2\tau_j/\gamma$ are consistent with the thermal picture. Moreover, in this limit, it can be easily seen from Eq. (6) that,

$$J_{\text{act}} = \frac{k(T_1^{\text{eff}} - T_N^{\text{eff}})}{2\gamma}\left[1 + \frac{mk}{2\gamma^2} - \frac{mk}{2\gamma^2}\sqrt{1 + \frac{4\gamma^2}{mk}}\right] + O(\tau_j^2), \tag{18}$$

which is the same as the well-known form of the thermal current [3, 5] to leading order in $\tau_1, \tau_N$. This can also be seen from Fig. 5(b) where $J_{\text{act}}$ converges to the effective thermal current for $\tau_1, \tau_N \ll \sqrt{m/k}$.

## 4 Conclusions

In summary, we have analytically studied the transport properties of a harmonic chain coupled to two active reservoirs which exert exponentially correlated stochastic forces on the boundary oscillators. We find that this active drive leads to a NESS carrying an energy current, which exhibits intriguing features like NDC and current reversal. For a simple model of dichotomous active force, we show that the negative differential conductivity results from a positive corre-

lation of the energy current and number of flips of the active force. The kinetic temperature profile, which, similar to the thermally driven scenario, remains uniform in the bulk of the system, also carries strong signatures of activity and an effective temperature picture cannot be consistently built.

Our work is the first to study the effect of active reservoirs on transport properties of extended systems. The results presented here are quite robust as the exponential correlation is a generic feature of active dynamics. However, signatures of specific dynamics are expected to be seen in the fluctuations of the current. It would be interesting to see if our results can be qualitatively verified in experiments with active reservoirs, say a collection of active Brownian particles [32], connected by passive polymers. Some other interesting questions are: What are the effects of disorder, anharmonicity and pinning in the presence of active driving? How do our results change, if the nonequilibrium reservoir is modeled by a chain of active particles in the spirit of [11, 50, 51]?

## Acknowledgements

The authors would like to thank Abhishek Dhar, Christian Maes and P. K. Mohanty for useful discussions.

**Funding information** U. B. acknowledges support from the Science and Engineering Research Board (SERB), India, under a Ramanujan Fellowship (Grant No. SB/S2/RJN-077/2018).

## A  Stationary state Current

In this section, we sketch the main steps of the computation of the current starting from Eq. (4) in the main text. For the sake of completeness we first rewrite the Langevin equations Eqs. (1),

$$M\ddot{X}(t) = -\Phi X(t) - \Gamma \dot{X}(t) + \Xi(t) + F(t), \tag{A.1}$$

where, $X(t) = \{x_l(t); l = 1,\ldots,N\}$ is a vector, $M$ is an $N$-dimensional diagonal matrix with $M_{lj} = m\delta_{l,j}$; $\Phi$ and $\Gamma$ are $N$-dimensional matrices given by

$$\begin{aligned} \Phi_{jl} &= k\left(2\delta_{j,l} - \delta_{j,l-1} - \delta_{j,l+1}\right), \\ \Gamma &= \Gamma_L + \Gamma_R, \quad \text{with} \quad (\Gamma_L)_{jl} = \gamma\delta_{j,1}\delta_{l,1}, \quad (\Gamma_R)_{jl} = \gamma\delta_{j,N}\delta_{l,N}. \end{aligned} \tag{A.2}$$

Moreover, the vectors $\Xi(t)$ and $F(t)$ represent the thermal and active forces exerted by the reservoirs on the boundary oscillators,

$$\Xi(t) = \Xi_L(t) + \Xi_R(t) \quad \text{with} \quad (\Xi_L)_j(t) = \xi_1(t)\delta_{j1} \quad \text{and} \quad (\Xi_R)_j(t) = \xi_N(t)\delta_{jN}, \tag{A.3a}$$

$$F(t) = F_L(t) + F_R(t) \quad \text{with} \quad (F_L)_j(t) = f_1(t)\delta_{j1} \quad \text{and} \quad (F_R)_j(t) = f_N(t)\delta_{jN}. \tag{A.3b}$$

Here, $\xi_{1,N}(t)$ are delta correlated white-noises, while the active noises $f_{1,N}(t)$ have an exponentially decaying auto-correlation,

$$\langle \xi_j(t)\xi_l(t')\rangle = \delta_{jl}\, 2\gamma T_j \delta(t-t'), \quad \text{and} \quad \langle f_j(t)f_l(t')\rangle = \delta_{jl}\, a_j^2 e^{-|t-t'|/\tau_j}. \tag{A.4}$$

Note that, even though Eq. (A.1) formally appears to be a limiting case of [11] with vanishing bulk activity, the two scenarios differ by their physical nature as well as emergent phenomena, as we will see below.

The stationary energy current flowing through the system can be expressed as $J = \langle \mathcal{J}(t) \rangle$ where

$$\mathcal{J}(t) = \dot{x}_1 [-\gamma \dot{x}_1 + \xi_1(t) + f_1(t)] \tag{A.5}$$

denotes the instantaneous work done by, the left reservoir on the left boundary oscillator and the statistical averaging is done over the stationary state. It is convenient to recast this energy current using the above matrix notation and separate it into two terms,

$$J = J_1 + J_2, \quad \text{with} \quad J_1 = -\text{Tr}\left[\langle \dot{X}(t)\dot{X}^T(t)\Gamma_L \rangle\right] \quad \text{and} \quad J_2 = \text{Tr}\left[\langle (\Xi_L + F_L)\dot{X}^T(t)\rangle\right], \tag{A.6}$$

where $\dot{X}^T$ denotes the transpose of the vector $\dot{X}$. In the following we compute $J_1$ and $J_2$ separately using the solution of Eq. (A.1),

$$X(t) = \int_{-\infty}^{\infty} \frac{d\omega}{2\pi} e^{-i\omega t} G(\omega)[\Xi(\omega) + F(\omega)], \tag{A.7}$$

where $G(\omega) = [-M\omega^2 + \Phi - i\omega(\Gamma_L + \Gamma_R)]^{-1}$ [see Eq. (10)]. Let us first consider,

$$J_1 = \int_{-\infty}^{\infty} \frac{d\omega}{2\pi} \int_{-\infty}^{\infty} \frac{d\omega'}{2\pi} \omega \omega' e^{-i(\omega+\omega')t} \text{Tr}\left[\langle \tilde{X}(\omega)\tilde{X}^T(\omega')\rangle \Gamma_L\right] \tag{A.8}$$

$$= \int_{-\infty}^{\infty} \frac{d\omega}{2\pi} \int_{-\infty}^{\infty} \frac{d\omega'}{2\pi} \omega \omega' e^{-i(\omega+\omega')t} \text{Tr}\left[G(\omega)\langle [\Xi(\omega)+F(\omega)][\Xi(\omega')+F(\omega')]\rangle G(\omega')\Gamma_L\right],$$

where we have used the fact that $G^T(\omega') = G(\omega')$ as $G$ is a symmetric matrix. The noise correlations appearing in the above equation can be evaluated in a straightforward manner using Eqs. (A.3)-(A.4). Since the noises from the two reservoirs are independent, it is natural to separate the corresponding contributions and write,

$$\langle [\Xi(\omega)+F(\omega)][\Xi(\omega')+F(\omega')]\rangle = 2\pi\delta(\omega+\omega')[S_L(\omega)+S_R(\omega)], \tag{A.9}$$

where the matrix elements of $S_{L,R}(\omega)$ are given by,

$$\left(S_L(\omega)\right)_{jl} = [2\gamma T_1 + \tilde{g}(\tau_1,\omega)]\delta_{j,1}\delta_{l,1}, \quad \text{and} \quad (S_R)_{jl} = [2\gamma T_N + \tilde{g}(\tau_N,\omega)]\delta_{j,N}\delta_{l,N}. \tag{A.10}$$

Here $\tilde{g}(\tau_j,\omega)$ denotes the Fourier transform of the active force auto-correlation

$$\tilde{g}(\tau_j,\omega) = a_j^2 \int_{-\infty}^{\infty} ds\, e^{i\omega s} e^{-|s|/\tau_j} = \frac{2a_j^2 \tau_j}{(1+\omega^2\tau_j^2)}. \tag{A.11}$$

Using Eqs. (A.9) and (A.10) in Eq. (A.8), we get,

$$J_1 = -\int_{-\infty}^{\infty} \frac{d\omega}{2\pi} \omega^2 \text{Tr}\left[G(\omega)\left(S_L(\omega)+S_R(\omega)\right)G^*(\omega)\Gamma_L\right], \tag{A.12}$$

where $G^*(\omega) = G(-\omega)$ denotes the complex conjugate of $G(\omega)$. Proceeding similarly for $J_2$, we have from Eq. (A.6) and Eq. (A.9),

$$J_2 = i\int_{-\infty}^{\infty} \frac{d\omega}{2\pi} \omega \text{Tr}[G^*(\omega)S_L(\omega)]. \tag{A.13}$$

Combining Eqs. (A.12) and (A.13) and rearranging the terms, we have,

$$J = \int_{-\infty}^{\infty} \frac{d\omega}{2\pi} \omega \text{Tr}\left[\left(iG^*(\omega)-\omega G^*(\omega)\Gamma_L G(\omega)\right)S_L(\omega)\right] - \int_{-\infty}^{\infty} \frac{d\omega}{2\pi} \omega^2 \text{Tr}\left[G^*(\omega)\Gamma_L S_R(\omega)\right]. \tag{A.14}$$

Now, remembering the definition of $G(\omega) = [-M\omega^2 + \Phi - i\omega(\Gamma_L + \Gamma_R)]^{-1}$, it can be easily shown that,

$$G^*(\omega)\Gamma_L G(\omega) = \frac{G(\omega) - G^*(\omega)}{2i\omega} - G^*(\omega)\Gamma_R G(\omega). \tag{A.15}$$

Using the above relation the first term of Eq. (A.14) can be further simplified,

$$\int_{-\infty}^{\infty} \frac{d\omega}{2\pi}\, \omega \, \mathrm{Tr}\left[\left(iG^*(\omega) - \omega G^*(\omega)\Gamma_L G(\omega)\right) S_L(\omega)\right]$$
$$= \frac{i}{2}\int_{-\infty}^{\infty} \frac{d\omega}{2\pi}\, \omega \, \mathrm{Tr}\left[\left(G(\omega) + G^*(\omega)\right) S_L(\omega)\right] + \int_{-\infty}^{\infty} \frac{d\omega}{2\pi}\, \omega^2 \, \mathrm{Tr}\left[G^*(\omega)\Gamma_R G(\omega) S_L(\omega)\right]. \tag{A.16}$$

The first integral on the second line vanishes as $\omega(G(\omega) + G^*(\omega))S_L(\omega)$ is an odd function of $\omega$, and we finally have, from Eqs. (A.14) and (A.16),

$$J = \int_{-\infty}^{\infty} \frac{d\omega}{2\pi}\, \omega^2 \, \mathrm{Tr}\left[G(\omega)\,\Gamma_R\, G^*(\omega)\, S_L(\omega) - G(\omega)\, S_R(\omega)\, G^*(\omega)\,\Gamma_L\right]. \tag{A.17}$$

From the expressions of $S_L(\omega)$ and $S_R(\omega)$ given in Eq. (A.10) it is immediately clear that $J$ separates into two parts — $J = J_{\mathrm{th}} + J_{\mathrm{act}}$, where,

$$J_{\mathrm{th}} = \gamma^2(T_1 - T_N)\int_{-\infty}^{\infty} \frac{d\omega}{2\pi}\, \omega^2 |G_{1N}(\omega)|^2,\quad \text{and} \tag{A.18a}$$

$$J_{\mathrm{act}} = J_{\mathrm{act}}^1 - J_{\mathrm{act}}^N,\quad \text{with}\quad J_{\mathrm{act}}^j = \gamma\int_{-\infty}^{\infty} \frac{d\omega}{2\pi}\, \omega^2 |G_{1N}(\omega)|^2 \tilde{g}(\tau_j, \omega). \tag{A.18b}$$

The thermal current $J_{\mathrm{th}}$ is well known in the literature [2,3] and is given by,

$$J_{\mathrm{th}} = \frac{k(T_1 - T_N)}{2\gamma}\left[1 + \frac{mk}{2\gamma^2} - \frac{mk}{2\gamma^2}\sqrt{1 + \frac{4\gamma^2}{mk}}\,\right]. \tag{A.19}$$

In the following we compute the active current $J_{\mathrm{act}}$ exactly. To this end, we first need the explicit form for the matrix element $G_{1N}(\omega)$. This has been calculated in the context of thermal transport [2], we revisit the calculation here for the sake of completeness.

By definition, $G(\omega)$ is the inverse of a tri-diagonal matrix (see Eq. (10)) and the elements $G_{ij}(\omega)$ can be computed explicitly exploiting this tridiagonal structure of $G^{-1}(\omega)$ [45]. In particular, we will need the following elements,

$$G_{l1}(\omega) = (-k)^{l-1}\frac{\theta_{N-l}}{\theta_N},\quad \text{and} \tag{A.20a}$$

$$G_{lN}(\omega) = (-k)^{N-l}\frac{\theta_{l-1}}{\theta_N}, \tag{A.20b}$$

where $\theta_l$ satisfies the recursion relation,

$$\theta_l = (-m\omega^2 + 2k)\,\theta_{l-1} - k^2\,\theta_{l-2}\quad \text{for } l = 2, 3, \ldots, N-1, \tag{A.21a}$$
$$\text{and}\quad \theta_N = (-m\omega^2 + 2k - i\omega\gamma)\,\theta_{N-1} - k^2\,\theta_{N-2}. \tag{A.21b}$$

Using the boundary conditions $\theta_0 = 1$ and $\theta_1 = (-m\omega^2 + 2k - i\omega\gamma)$ [45], the recursion relation (A.21a) can be solved in a straightforward manner. It is convenient to express the solution as,

$$\theta_l = \frac{(-k)^{l-1}}{\sin(q)}\left[k\sin((l+1)q) - i\omega\gamma\sin(lq)\right]\quad \text{for } l = 2, 3, \ldots, N-1, \tag{A.22}$$



where $q$ and $\omega$ are related by,

$$\cos q = \left(1 - \frac{m\omega^2}{2k}\right) \quad \Rightarrow \quad \omega = \omega_c \sin\frac{q}{2}, \tag{A.23}$$

where $\omega_c = 2\sqrt{k/m}$. Using Eq. (A.22) in Eq. (A.21b) we then have,

$$\theta_N = \frac{(-k)^N}{\sin q}\left[a(q)\sin(Nq) + b(q)\cos(Nq)\right], \tag{A.24}$$

where,

$$a(q) = -\frac{2i\gamma\omega}{k} + \cos q\left(1 - \frac{\gamma^2\omega^2}{k^2}\right), \quad \text{and} \quad b(q) = \sin q\left(1 + \frac{\gamma^2\omega^2}{k^2}\right). \tag{A.25}$$

Now we can proceed to compute the active current. Using Eq. (A.24) and Eq. (A.20b) for $l = 1$ in Eq. (A.18b), we get,

$$J_{\text{act}}^1 = \frac{\gamma}{\pi k^2}\int_0^\infty d\omega\,\omega^2\frac{\sin^2 q}{|a(q)\sin(Nq) + b(q)\cos(Nq)|^2}\tilde{g}(\tau_1,\omega). \tag{A.26}$$

At this point, it is important to note that, for $\omega > \omega_c$, $q$ becomes complex. Thus, for large $N$, in the region $\omega > \omega_c$, the integrand vanishes exponentially as $\exp(-2N\bar{q})$, where $\bar{q}$ is real. Thus, to compute the current for thermodynamically large systems, we can limit the range of integration in Eq. (A.26) to be $0 \le \omega \le \omega_c$ or equivalently, $0 \le q \le \pi$. Moreover, the functions $\sin(Nq)$ and $\cos(Nq)$ are highly oscillatory for large $N$ and in the $N \to \infty$ limit, we can average over $x = Nq$ and write [10],

$$J_{\text{act}}^1 = \frac{\gamma}{\pi k^2}\int_0^{\omega_c} d\omega\,\omega^2\sin^2 q\,\tilde{g}(\tau_1,\omega)\int_0^{2\pi}\frac{dx}{2\pi}\frac{1}{|a(q)\sin x + b(q)\cos x|^2}. \tag{A.27}$$

The $x$-integral has a simple form and can be evaluated exactly (see Sec. 2.558 in [52]),

$$\int_0^{2\pi}\frac{dx}{2\pi}\frac{1}{(c_1\sin x + d\cos x)^2 + c_2^2\sin^2 x} = -\frac{1}{dc_2}, \tag{A.28}$$

where we have denoted $c_1 = \text{Re}[a(q)]$, $c_2 = \text{Im}[a(q)]$ and $d = b(q)$ for notational simplicity. Substituting Eq. (A.28) in Eq. (A.27), we get,

$$J_{\text{act}}^1 = \frac{k}{2\pi}\int_0^{\omega_c} d\omega\,\frac{\omega\sin q}{k^2 + \gamma^2\omega^2}\tilde{g}(\tau_1,\omega) = \frac{k}{2}\int_0^\pi\frac{dq}{\pi}\left|\frac{d\omega}{dq}\right|\frac{\omega\sin q}{k^2 + \gamma^2\omega^2}\tilde{g}(\tau_1,\omega). \tag{A.29}$$

Thereafter, using the Jacobian $\left|\frac{d\omega}{dq}\right| = \frac{k\sin q}{m\omega}$, we arrive at,

$$J_{\text{act}}^1 = \int_0^\pi\frac{dq}{\pi}\frac{mk\tau_1 a_1^2\sin^2 q}{[mk + 2\gamma^2(1-\cos q)][m + 2k\tau_1^2(1-\cos q)]}, \tag{A.30}$$

where we have also expressed $\tilde{g}(\tau_1,\omega)$ as a function of $q$. This integral can be evaluated exactly and leads to,

$$J_{\text{act}}^1 = \frac{m}{2\gamma^2}a_1^2\mathcal{E}_1 \quad \text{with} \quad \mathcal{E}_1 = \frac{\tau_1^2 k^2\left[\sqrt{1 + \frac{4\gamma^2}{mk}} - 1\right] + \gamma^2\left[1 - \sqrt{1 + \frac{4k\tau_1^2}{m}}\right]}{2\tau_1(\tau_1^2 k^2 - \gamma^2)}. \tag{A.31}$$

One can similarly obtain $J_{\text{act}}^N = \frac{m}{2\gamma^2} a_N^2 \mathcal{E}_N$, where,

$$\mathcal{E}_N = \frac{\tau_N^2 k^2 \left[ \sqrt{1 + \frac{4\gamma^2}{mk}} - 1 \right] + \gamma^2 \left[ 1 - \sqrt{1 + \frac{4k\tau_N^2}{m}} \right]}{2\tau_N(\tau_N^2 k^2 - \gamma^2)}. \tag{A.32}$$

The total active current is obtained by combining Eq. (A.32) and (A.31), which is quoted in Eq. (5).

# B  Kinetic temperature profile

The kinetic temperature of the $l^{th}$ oscillator as defined in the main text is given by,

$$\hat{T}_l = m \langle \dot{x}_l^2(t) \rangle . \tag{B.1}$$

Since we are primarily interested in the effect of the active driving, we put $T_1 = T_N = 0$. Then, from Eq. (A.7), we get,

$$\hat{T}_l = m \int_{-\infty}^{\infty} \frac{d\omega}{2\pi} \omega^2 \Big[ |G_{l1}(\omega)|^2 \tilde{g}(\tau_1, \omega) + |G_{lN}(\omega)|^2 \tilde{g}(\tau_N, \omega) \Big]. \tag{B.2}$$

From Eqs. (A.20) we have,

$$|G_{l1}(\omega)|^2 = \frac{k^{2N-4}}{\sin^2 q |\theta_N|^2} \Big| k \sin(N-l+1)q - i\omega\gamma \sin(N-l)q \Big|^2, \tag{B.3a}$$

$$|G_{lN}(\omega)|^2 = \frac{k^{2N-4}}{2 \sin^2 q |\theta_N|^2} \Big| k \sin(lq) - i\omega\gamma \sin(l-1)q \Big|^2. \tag{B.3b}$$

We are particularly interested in the behavior of the kinetic temperature in the bulk in the thermodynamic limit $N \to \infty$. For this purpose we evaluate $\hat{T}_l$ for $l = N/2 + \ell$ where $\ell \ll N$. Let us first consider the contribution from the left reservoir, i.e., the first term in Eq. (B.2). Once again, the integrand vanishes exponentially for $\omega > \omega_c$ in the large $N$ limit, and we can write,

$$\begin{aligned} I_1 &\equiv \int_{-\infty}^{\infty} \frac{d\omega}{2\pi} \omega^2 |G_{l1}(\omega)|^2 \tilde{g}(\tau_1, \omega) \\ &= \frac{1}{k^4} \int_0^{\pi} \frac{dq}{2\pi} \left| \frac{d\omega}{dq} \right| \omega^2 \frac{k^2(1 - \cos(N-2\ell+2)q) + \omega^2\gamma^2(1 - \cos(N-2\ell)q)}{|a(q)\sin(Nq) + b(q)\cos(Nq)|^2} \tilde{g}(\tau_1, \omega). \end{aligned} \tag{B.4}$$

As before, in the $N \to \infty$ limit, we can average over the fast oscillations in $x = Nq$. For this purpose, let us note,

$$\int_0^{2\pi} \frac{dx}{2\pi} \frac{\sin x}{(c_1 \sin x + d \cos x)^2 + c_2^2 \sin^2 x} = \int_0^{2\pi} \frac{dx}{2\pi} \frac{\cos x}{(c_1 \sin x + d \cos x)^2 + c_2^2 \sin^2 x} = 0 . \tag{B.5}$$

Using these identities and Eq. (A.28), Eq. (B.4) reduces to,

$$I_1 = \frac{1}{2\gamma k} \int_0^{\pi} \frac{dq}{2\pi} \left| \frac{d\omega}{dq} \right| \frac{\omega}{\sin q} \tilde{g}(\tau_1, \omega) = \frac{1}{4\pi\gamma m} \int_0^{\pi} dq \, \tilde{g}(\tau_1, \omega(q)) . \tag{B.6}$$

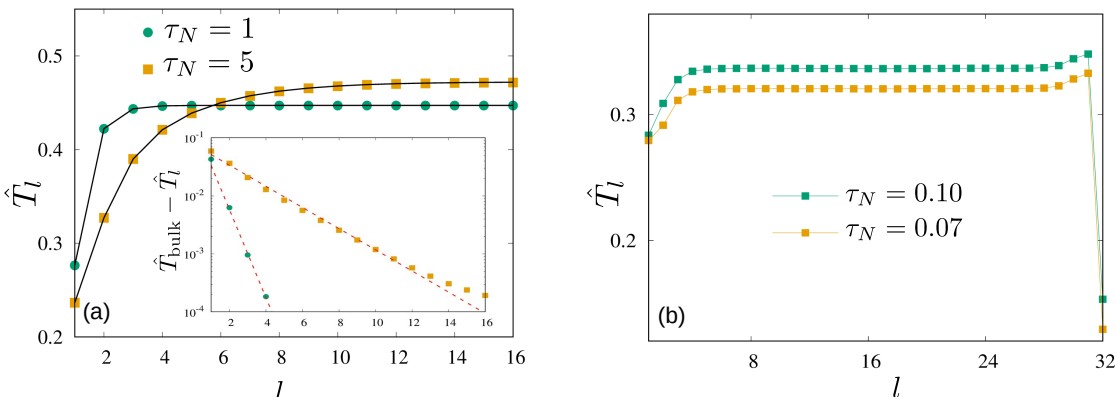

Figure 6: Boundary layer properties of the kinetic temperature profile: (a) shows $\hat{T}_l$ profile near the left boundary. The main plot compares the contributions from Eq. (B.13), (B.18) and (B.21) (in solid black lines) with numerical simulations (in colored symbols). The inset plot shows the exponential decay of $\hat{T}_l$ from $\hat{T}_{\text{bulk}}$ at the left boundary for a system of $N = 64$ oscillators with $m = 1$, $k = 1$, $\gamma = 1$ and $\tau_1 = 1$. (b) shows the absence (presence) of the boundary kinks when $\tau_{1,N}$ is much larger (smaller) than $\omega_c = 2\sqrt{k/m}$. Here $N = 32$ with $m = 1$, $k = 0.5$, $\gamma = 1$ and $\tau_1 = 1$.

The $q$-integral can be evaluated exactly, and yields,

$$I_1 = \frac{1}{2\gamma m} \frac{a_1^2 \tau_1}{\sqrt{1 + 4\tau_1^2 k/m}} \,. \tag{B.7}$$

The integral involving $G_{lN}$ can also be performed following the same procedure and results in,

$$I_2 \equiv \int_{-\infty}^{\infty} \frac{d\omega}{2\pi} \omega^2 |G_{lN}(\omega)|^2 \tilde{g}(\tau_N, \omega) = \frac{1}{2\gamma m} \frac{a_N^2 \tau_N}{\sqrt{1 + 4\tau_N^2 k/m}} \,. \tag{B.8}$$

Combining these results, we see that the kinetic temperature remains uniform in the bulk and is given by,

$$\hat{T}_{bulk} = \frac{1}{2\gamma} \left( \frac{a_1^2 \tau_1}{\sqrt{1 + 4\tau_1^2 k/m}} + \frac{a_N^2 \tau_N}{\sqrt{1 + 4\tau_N^2 k/m}} \right) \,. \tag{B.9}$$

This is the result presented in Eq. (6).

For a finite chain the kinetic temperature deviates from $\hat{T}_{bulk}$ near the boundaries giving rise to exponentially decaying boundary layers; see Fig. 6(a). To obtain the behavior of the boundary layers, we need to evaluate Eq. (B.2) in the limits $l \ll N$ and $l \sim N$.

## B.1 $\hat{T}_l$ near left boundary

Let us first concentrate near the left boundary, where $l = 1, 2, 3 \ldots \ll N$. For convenience, we rewrite Eq. (B.2) as,

$$\hat{T}_l = m[L_1(l, \tau_1) + L_N(l, \tau_N)], \tag{B.10}$$

where $L_1(l, \tau_1)$ and $L_N(l, \tau_N)$ denote the contributions from the left and right reservoirs respectively,

$$L_1(l, \tau) = \int_{-\infty}^{\infty} \frac{d\omega}{2\pi} \omega^2 |G_{l1}(\omega)|^2 \tilde{g}(\tau, \omega), \tag{B.11a}$$

$$L_N(l, \tau) = \int_{-\infty}^{\infty} \frac{d\omega}{2\pi} \omega^2 |G_{lN}(\omega)|^2 \tilde{g}(\tau, \omega). \tag{B.11b}$$

We first evaluate the contribution from the right reservoir $L_N(l, \tau)$. In this case, once again, the contribution coming from $|\omega| > \omega_c$ vanishes exponentially for large $N$ and in the thermodynamic limit Eq. (B.11b) reduces to,

$$L_N(l, \tau) = \frac{1}{k^4} \int_0^{\omega_c} \frac{d\omega}{\pi} \omega^2 \frac{k^2 \sin^2(lq) + \omega^2 \gamma^2 \sin^2(l-1)q}{|a(q)\sin(Nq) + b(q)\cos(Nq)|^2} \tilde{g}(\tau, \omega). \tag{B.12}$$

Averaging over the fast oscillations in the $N \to \infty$ limit and using Eq. (A.28), we get,

$$L_N(l, \tau) = \frac{1}{2\gamma m} \int_0^{\pi} \frac{dq}{\pi} \frac{k^2 \sin^2(lq) + \omega^2 \gamma^2 \sin^2(l-1)q}{(k^2 + \gamma^2 \omega^2)} \tilde{g}(\tau, \omega). \tag{B.13}$$

Though this integral does not yield any closed form expression, it can be evaluated numerically for arbitrary $l$ and $\tau$.

Next, we consider the contribution from the left reservoir $L_1(l, \tau)$. It turns out that $L_1(l, \tau)$ (Eq. (B.11a)) has non-vanishing contribution from both $|\omega| < \omega_c$ and $|\omega| > \omega_c$. Thus, it is convenient to rewrite Eq. (B.11a) as,

$$L_1(l, \tau) = L_1^b(l, \tau) + L_1^o(l, \tau), \tag{B.14}$$

where $L_1^b(l, \tau)$ and $L_1^o(l, \tau)$ denote the contributions from $|\omega| < \omega_c$ and $|\omega| > \omega_c$ respectively. For $|\omega| > \omega_c$, Eq. (A.23) implies that $q = \pi - i\bar{q}$, where $\bar{q}$ is real. We first evaluate the contribution from this region,

$$\begin{aligned} L_1^o(\tau) &= \int_{\omega_c}^{\infty} \frac{d\omega}{\pi} \omega^2 |G_{l1}(\omega)|^2 \tilde{g}(\tau, \omega) \\ &= \frac{1}{k^4} \int_{\omega_c}^{\infty} \frac{d\omega}{\pi} \omega^2 \frac{|ik \sinh(N-l+1)\bar{q} - \omega\gamma \sinh(N-l)\bar{q}|^2}{|ia(\bar{q})\sinh(Nq) - b(\bar{q})\cosh(N\bar{q})|^2} \tilde{g}(\tau, \omega), \end{aligned} \tag{B.15}$$

where we have used the identities,

$$\sin(nq) = (-1)^{n+1} i \sinh(n\bar{q}), \quad \text{and} \quad \cos(nq) = (-1)^n \cosh(n\bar{q}), \quad n = 0, 1, 2, \dots. \tag{B.16}$$

In the $N \to \infty$ limit, Eq. (B.15) reduces to

$$L_1^o(\tau) = \frac{1}{k^4} \int_{\omega_c}^{\infty} \frac{d\omega}{\pi} \omega^2 e^{-2l\bar{q}} \frac{k^2 e^{2\bar{q}} + \omega^2 \gamma^2}{|ia(\bar{q}) - b(\bar{q})|^2} \tilde{g}(\tau, \omega). \tag{B.17}$$

The integral over $\omega \in [\omega_c, \infty]$ can be converted to an integral over $\bar{q} \in [0, \infty]$ using the relation $\omega = \omega_c \cosh(\bar{q}/2)$ [see Eq. (A.23)], to get,

$$L_1^o(\tau) = \frac{\omega_c^3}{2} \int_0^{\infty} \frac{d\bar{q}}{\pi} e^{-2l\bar{q}} \frac{\sinh(\bar{q}/2) \cosh^2(\bar{q}/2)}{k^2 + \gamma^2 \omega_c^2 \cosh^2(\bar{q}/2) e^{-2\bar{q}}} \tilde{g}\left(\tau, \omega_c \cosh \frac{\bar{q}}{2}\right). \tag{B.18}$$

This, again, can be evaluated numerically for arbitrary $l$. For $|\omega| < \omega_c$, $q$ is real and the contribution to Eq. (B.11a) is given by,

$$
\begin{aligned}
L_1^b(\tau) &= \int_0^{\omega_c} \frac{d\omega}{\pi} \omega^2 |G_{l1}(\omega)|^2 \, \tilde{g}(\tau, \omega) \\
&= \frac{1}{k^4} \int_0^{\omega_c} \frac{d\omega}{\pi} \omega^2 \frac{k^2 \sin^2(N-l+1)q + \omega^2 \gamma^2 \sin^2(N-l)q}{|a(q)\sin(Nq) + b(q)\sin(Nq)|^2} \, \tilde{g}(\tau, \omega).
\end{aligned}
\tag{B.19}
$$

For $l \ll N$ and $N \to \infty$ limit, averaging over the fast oscillations $x = Nq$ involves integrals of the form,

$$
\begin{aligned}
Q(v) &= \int_0^{2\pi} \frac{dx}{2\pi} \frac{\sin^2(x-v)}{(c_1 \sin x + d \cos x)^2 + c_2^2 \sin^2 x} \\
&= \frac{\left[c_1^2 + c_2^2 - d^2\right]\cos(2v) - \left[c_1^2 + (c_2-d)^2 + 2c_1 d \sin(2v)\right]}{2dc_2\left[c_1^2 + (c_2-d)^2\right]},
\end{aligned}
\tag{B.20}
$$

where $v > 0$ is arbitrary and $c_1 = \text{Re}[a(q)]$, $c_2 = \text{Im}[a(q)]$ and $d = b(q)$ as before. Using the above result in Eq. (B.19) with appropriate values of $v$,

$$
\begin{aligned}
L_1^b(\tau) &= \frac{1}{k^4} \int_0^{\omega_c} \frac{d\omega}{\pi} \omega^2 \left[k^2 Q((l-1)q) + \omega^2 \gamma^2 Q(lq)\right] \tilde{g}(\tau, \omega) \\
&= \frac{\omega_c^3}{2k^4} \int_0^{\pi} \frac{d\bar{q}}{\pi} \sin^2 \frac{q}{2} \cos \frac{q}{2} \left[k^2 Q((l-1)q) + \omega_c^2 \sin^2 \frac{q}{2} \gamma^2 Q(lq)\right] \tilde{g}\left(\tau, \omega_c \sin \frac{q}{2}\right).
\end{aligned}
\tag{B.21}
$$

Adding the contributions given by Eq. (B.13), (B.18) and (B.21), we can evaluate the kinetic temperature profile near the left boundary, which is shown in Fig. 6.

## B.2   $\hat{T}_l$ near right boundary

The behavior near the right boundary can be obtained in a similar manner. For this purpose, it is convenient to define, $\ell = N - l + 1$, such that $\ell = 1, 2, 3, \ldots \ll N$ corresponds to the oscillators near the right boundary. Next, we note that, from Eqs. (B.3a) and (B.3b),

$$
|G_{l1}(\omega)|^2 = |G_{N-l+1,N}(\omega)|^2,
\tag{B.22}
$$
$$
|G_{lN}(\omega)|^2 = |G_{N-l+1,1}(\omega)|^2.
\tag{B.23}
$$

Then the $\hat{T}_l$ profile near the right boundary ($l \sim N$) is given by,

$$
\begin{aligned}
\hat{T}_{N-\ell+1} &= m \int_{-\infty}^{\infty} \frac{d\omega}{2\pi} \omega^2 \left[|G_{N-\ell+1,1}(\omega)|^2 \tilde{g}(\tau_1, \omega) + |G_{N-\ell+1,N}(\omega)|^2 \tilde{g}(\tau_N, \omega)\right] \\
&= m \int_{-\infty}^{\infty} \frac{d\omega}{2\pi} \omega^2 \left[|G_{\ell,N}(\omega)|^2 \tilde{g}(\tau_1, \omega) + |G_{\ell,1}(\omega)|^2 \tilde{g}(\tau_N, \omega)\right] \\
&= m\left[L_N(\ell, \tau_1) + L_1(\ell, \tau_N)\right],
\end{aligned}
\tag{B.24}
$$

where $L_1(\ell, \tau)$ and $L_N(\ell, \tau)$ are obtained from Eqs. (B.13), (B.18) and (B.21).

Interestingly, boundary kinks, which are absent in the active regime appear in the passive limit, similar to the thermal scenario. This is shown in Fig. 6(b) where kinks are visible near the right boundary as the activity of the corresponding reservoirs is small, whereas no kinks are visible near the left reservoir, which remains in the strongly active regime.

# C  Linear response: Differential conductivity

In this section we derive an expression for the differential conductivity for the energy current using nonequilibrium response theory. The nonequilibrium linear response relations are most conveniently derived using a trajectory based approach and differ from the equilibrium response by the presence of an additional 'frenetic' contribution, which is symmetric under time-reversal [47].

To derive linear response relations for the activity driven chain using the trajectory based approach, a specific dynamics of the active force is required. Here we take the specific example of dichotomous active forces which reverse their directions with rates $\alpha_1$ and $\alpha_N$ at the two reservoirs [see Eq. (3)]. In this case, it is most natural to consider a perturbation $\alpha_j \to \alpha_j + d\alpha_j$ and express the differential conductivity as,

$$\chi_j \equiv \frac{dJ_{\mathrm{act}}}{d\tau_j} = -2\alpha_j^2 \frac{dJ_{\mathrm{act}}}{d\alpha_j}, \tag{C.1}$$

where we have used the fact that $\tau_j = 1/(2\alpha_j)$ in this scenario. Let $\omega = \{x_i(s), \sigma_1(s), \sigma_N(s); 0 \le s \le t\}$ denote a trajectory of the system during the interval $[0, t]$ and $P_{\alpha_1, \alpha_N}(\omega)$ denote the corresponding probability. Of course, the trajectory probability depends on the various system parameters, but since we are interested in the response to a change in the flip rate, it suffices to consider the $\alpha_j$ dependence. Hence, we can write,

$$P_{\alpha_1, \alpha_N}(\omega) = e^{-(\alpha_1 + \alpha_N)t} \alpha_1^{n_1} \alpha_N^{n_N} \mathcal{U}(\omega), \tag{C.2}$$

where $n_1$ and $n_N$ denote the number of flips of the active force at the left and right boundaries during time $[0, t]$ and $\mathcal{U}(\omega)$ contains the $x_j$-dependent components. The weight of the same trajectory $\omega$ changes upon adding the perturbation i.e., say, changing $\alpha_1 \to \alpha_1 + d\alpha_1$. Note that this change does not affect $\mathcal{U}(\omega)$. The linear response of the expectation value of any observable $\langle O \rangle$ to this change can be expressed as a connected correlation in the unperturbed state [47],

$$\frac{d\langle O \rangle}{d\alpha_1} = -\langle \mathcal{A}(\omega); O(\omega) \rangle, \tag{C.3}$$

where $\mathcal{A}(\omega) = -\frac{d}{d\alpha_1} \log P_{\alpha_1, \alpha_N}(\omega)$ is the excess action associated to the trajectory $\omega$ due to the perturbation.

Then, from Eq. (C.2), we have, for the active current $J_{\mathrm{act}} = \langle \mathcal{J}(t) \rangle$,

$$\frac{dJ_{\mathrm{act}}}{d\alpha_1} = \frac{1}{\alpha_1} [\langle n_1 \mathcal{J}(t) \rangle - \langle n_1 \rangle \langle \mathcal{J}(t) \rangle]]. \tag{C.4}$$

The response to a change in the right reservoir is also given by a similar expression. The stationary response is obtained by taking the $t \to \infty$ limit which is quoted in Eq. (15), in terms of the activity parameter $\tau_j = 1/(2\alpha_j)$. Figure 7 compares the exact analytical response $\frac{dJ_{\mathrm{act}}}{d\alpha_1}$ obtained from Eqs. (A.31)-(A.32) using $\alpha_1 = 1/(2\tau_1)$ with the prediction (C.4), measured from numerical simulations, which shows an excellent match. As observed from the figure, that the susceptibility becomes negative beyond a certain $\alpha_1$, which depends on $k$ (and also other parameters of the system).

We close this discussion with a final remark. The dynamics of the active noise $f_j = a_j \sigma_j$ is symmetric under time-reversal if $f_j$ is considered as a force and hence, the nonequilibirum linear response obtained in Eq. (C.4) is purely frenetic. Frenetic contribution to the linear response is known to result in negative differential response in various contexts [36]. The activity driven harmonic chain provides another example where the same mechanism works, although the absence of any equilibrium limit and the nature of the perturbation here means that there is no traditional regime where one recovers a Kubo-like formula.

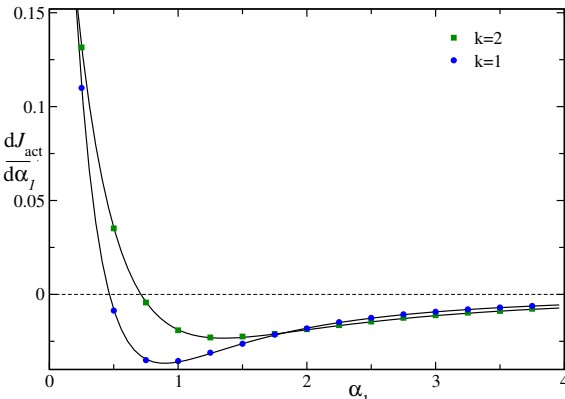

Figure 7: Plot of $\frac{dJ_{\text{act}}}{d\alpha_1}$ versus $\alpha_1$ for the dichotomous active noise for two different values of $k$: the solid lines show the exact expression obtained from Eq. (A.31) while the symbols show the predicted correlation [see Eq. (C.4)] measured from numerical simulations.

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
