# Peer review of "Activity driven transport in harmonic chains"

_SciPost Physics, doi:SciPost Phys. 13, 041 (2022)_

## Round 1 · Referee Report · Anonymous (Referee 1) · 2022-6-28

Strengths

  • novel model of reservoirs
  • clearly written

Report

The manuscript deals with a model of transport in a linear
chain driven by nonequilibrium (thermal and
nonthermal) energy baths. The main novelty of the work is the introduction
of active reservoirs, an issue that has not been considered before in
this field. The authors demonstrate two effects
(i) negative differential conductivity (NDC), and (ii) current direction
reversal at some finite value of the activity drive.
Although (i) has been reported in many examples with standard thermal
baths, (ii) is a novel effect characteristic of the non-equilibrium
nature of the reservoirs. I think the work may open a novel
research subfield (for instance the effect of acive baths in
nonlinear systems).

The results are obtained via the well-known approach of nonequilibrium
Green functions, a method that, with little extension of the calculation
(namely addition of the active forces in eq. (9)) allows the authors to
derive expression of the main observables (energy current and temperature
field).

The main idea of the paper is interesting and original. The methods
are sound. The presentation
is very accessible and self-contained (the Authors wisely condensed the technical
details in the appendix, leaving only the results in the main text).

I would recommend publication after the following issues have been
considered and the text revised accordingly.

Requested changes

1- I do not quite agree with the sentence "Equation (12) is a Landauer-like formula where the transmission coefficient depends on ...". The transmission coefficient is an intrisic property of the chain (and the boundary conditions). What is different here ar just the source terms g(tau,omega). As a related remark, after eq. (12): upon comparison with the Landauer formalua, "phonon spectrum $\omega^2 G_{1N}(\omega)$" should rather be "transmission coefficient $\omega^2 |G_{1N}(\omega)|^2$" (notice the square missing)

2- same in the caption of Figure 3: I would not call the curves a plot of the "phonon band" but the "transmission coefficient".

3- to demonstrate NDC Authors use an approach based on eq.(14) and comment that "when the number ... is positively correlated with the current an NDC emerges". However this is not very explicative: how can we understand when this happens in (14)? this point need some clarification (perhaps upon extending the comments at the end of appendix C?).

---

## Round 1 · Referee Report · Anonymous (Referee 2) · 2022-7-6

Report

Warnings issued while processing user-supplied markup:

  • Inconsistency: plain/Markdown and reStructuredText syntaxes are mixed. Markdown will be used.
    Add "#coerce:reST" or "#coerce:plain" as the first line of your text to force reStructuredText or no markup.
    You may also contact the helpdesk if the formatting is incorrect and you are unable to edit your text.

In this paper, the authors study a harmonic chain coupled to an active bath. More precisely, the system is subjected at its two extremities, i.e. at site $j=1$ and site $j=N$, to a noise which has a non-zero persistence time, respectively $\tau_1$ and $\tau_N$ (and thus does not satisfy the fluctuation-dissipation theorem), in addition to a Gaussian white noise (which satisfies the fluctuation-dissipation theorem). There is no other source of noise in the system. They show analytically that the system reaches a nonequilibrium steady state, with a nonzero energy current. The main focus of the paper is on the characterisation of this current, which exhibits two interesting features: (i) a non-monotonic behavior as a function of the persistence times $\tau_1$ and $\tau_N$ (which they call negative differential conductivity'') and (ii) a change of sign again as $\tau_1$ and $\tau_N$ are varied (which they callcurrent reversal''). These two effects do not have any counterpart in the case where the two baths are passive, i.e., when the noise at the two extremities is purely thermal, a case that was studied a long time ago by Rieder, Lieb and Lebowitz.

The paper is physically sound and presents one rare example where the non-equilibrum dynamics of an extended system coupled to an active bath can be carried out analytically. As far as I know, these results are new. Most of them are also carefully compared to numerical simulations, showing a very good agreement. The organisation of the paper as well as the presentation of the results is rather clear -- although the English wording still requires some work (see below) -- while technical details have been relegated to three Appendices.

Therefore, I would like to recommend the publication of the present manuscript to SciPost. The authors might however consider the following comments:

i) Below Eq. (2): please indicate that the indices $j$ and $l$ take only the values $j,l =1,N$.

ii) In Eq. (5) the authors mention both the thermal and active currents, but eventually give only the active one. I think that they should at least refer to Eq. (11) where the thermal current is given.

iii) In the caption of Fig. 2, the authors should describe what $\tau_m$ is (and maybe define this notation also in subsection 3.1.1)

iv) Above Eq. (13) the authors mention the tri-diagonal structure of the matrix $G(\omega)$. They should at least refer to Eq. (35) in Appendix A. In fact, since this matrix plays an important role in the computation, I would suggest to give it in the text earlier -- maybe around Eq. (8).

v) In Eq. (13), recall that $j=1$ or $j=N$.

vi) In the caption of Fig. 4, as well as in Section 3.1.2, the notation $(\bar{\tau}, \bar{\tau})$ is a bit strange: maybe choose a better one?

vii) On p. 8, below Eq. (15): what do the authors mean by "... numerical simulations performed with Eq. (3) in Fig. 5(a)"? They should clarify their statement here.

viii) In Appendix A, below Eq. (23) the authors use the fact that the matrix $G(\omega)$ is symmetric but this matrix is given only later in the paper, in Eq. (35). As mentioned above, I think that this matrix should be given much earlier (and in the text, not only in the Appendix).

ix) In Eq. (25) there is a misprint: $H_R$ should be $S_R$.

x) Below Eq. (31): "The fist term on the second line..." --> "The first term in the integrand on the second line"... Without this precision, the sentence sounds a bit odd.

xi) In Appendix A, below Eq. (41): "given by" has nothing to do here. It should be deleted.

xii) Above Eq. (44) the authors could give a hint or a reference on how to evaluate this integral.

xiii) At the end of Appendix C, the authors write that the linear response is purely "frenetic". They should at least explain a bit what is meant by "frenetic" here since here this comes completely out of the blue.

Minor points

As mentioned above, the paper needs some polishing. Here is a list (surely non-exhaustive) of instances where the wording needs to be corrected:

1) p .2: "on study of simple..." --> "on THE study of simple..."

2) p. 2: "introducing disorders" --> "introducing disorder"

3) p. 2: "lies with the Lorentzian.." --> "lies IN the Lorentzian.."

4) p. 2: "a uniform value at the bulk" --> "a uniform value IN the bulk"

5) p. 3, below Eq. (3): "as such exponential correlations" --> "SINCE exponential correlations"

6) On top of p. 4: "the average is over the NESS" --> "the average is taken in the NESS"

7) Below Eq. (6): "This suggests the possibility of interpreting..." --> "This would suggest..." since eventually, this interpretation is seemingly wrong.

8) Below Eq. (6): "different than thermal ones" --> "different FROM thermal ones". This mistake also appears in some other places in the paper.

9) On p. 5 below Eq. (11): "remains same" --> "remains THE same"

10) On p. 5: "is obtained exploiting" --> "is obtained BY exploiting"

11) On p. 8: above Eq. (15): "the the kinetic" --> "the kinetic"

12) On p. 8: "remain different than" --> "remainS different FROM"

13) On p. 8: "different than the energy" --> "different FROM the energy"

14) On p. 9, the expression for $J_{\rm act}$ in the passive limit does not have any number, please add one.

15) Below that un-numbered equation: "which is same" --> "which is THE same"

16) On p. 9, in the conclusion: "leads to an NESS" --> "leads to A NESS"

17) In Appendix A, Eq. (22) please replace the full stop by a coma.

18) Similarly, in Appendix A, Eq. (25) please replace the full stop by a coma.

19) In Appendix B, the first pair of equations is numbered as (51) and (52) while in many other places, they used (33a) and (33b) or (36a) and (36b). Pease uniformise the notations.

20) Below Eq. (52): "at the bulk" --> "IN the bulk"

---

## Round 2 · Referee Report · Anonymous (Referee 2) · 2022-7-13

Report

In this revised version, the authors have satisfactorily taken into account my comments and made the appropriate modifications. Therefore I can recommend the publication of this manuscript in SciPost. I just noticed a small typo in the paragraph which the authors have added in Appendix C: below Eq. (77), "that the susceptibility" --> "the susceptibility".

---

## Round 2 · Referee Report · Anonymous (Referee 1) · 2022-7-13

Report

The Authors revised the manuscript in a satisfactory way. I recommend publication in Scipost Physics in the present form.

---

## Round 2 · Author Response

Dear Editor,

Thank you for forwarding the reports and giving us the opportunity to resubmit our article in Scipost Physics. We also thank both the Referees for their careful reading of the manuscript and the constructive reports. We are glad to see that both the Referees appreciate the scientific merit of our article and recommend publication.

We hereby resubmit a modified version of our article where we have implemented the suggestions of the Referees. Detailed reply to both the reports are appended below. A list of changes is also provided; major modifications are also highlighted in color in the manuscript.

We hope that the current version is suitable for publication in SciPost Physics.

Sincerely, Ion Santra and Urna Basu.

Reply to Anonymous Report 1

We thank the Referee for the positive report and the valuable suggestions, which we have implemented in the revised version.

1 and 2- We have changed 'phonon spectrum' to 'phonon transmission coefficient' in all the relevant places. We also thank the Referee for pointing out the typo, which we have now corrected.

3- We have now added a few sentences in the beginning and at the end of Appendix C, clarifying the nonequilibrium linear response formalism. The response formula tells us that the susceptibility is negative when the <n J> correlation [Eq. (15) in current version] is positive. However, we do not have any clear physical understanding as to when this occurs.

Reply to Anonymous Report 2

We thank the Referee for appreciating our manuscript and for providing a detailed and constructive report. All the changes suggested in the comments (i)--(xii) have been implemented. (xiii) We have added a few sentences in the beginning and at the end of Appendix C clarifying the query.

Minor points: All the typos noted by the Referee and a few additional ones, which we found while rereading, have been corrected.

---

## Round 2 · List of Changes

1. We have added 'where j,l=1,N' in Eq. (2).

  2. We have referred to Eq. (14) just before Eq. (5).

  3. We have given the explicit tri-diagonal form of G(w) in Eq. (9).

  4. We have explained the notation \tau_m in the caption of Figure 2 and Sec. 3.1.1.

  5. We have consistently changed 'phonon spectrum' to 'phonon transmission coefficient'. In particular, the sentence refereeing to the Landauer like formula has been modified.

  6. We have now referred to the saddle point as \tau_1=\tau_N=\bar \tau

  7. The sentence refereeing Fig. 5(a) has been modified.

  8. We have changed the sentence after Eq. (33) to "The first integral on the second line vanishes as ..."

  9. The beginning and end of Appendix C has been modified, clarifying nonequilibrium linear response formalism.

  10. We have corrected all the typos pointed out by the Referees and a few additional ones.

All the significant changes have been marked in red.

---

## Editorial Decision

published